# Reduced Retinal Blood Vessel Densities Measured by Optical Coherence Tomography Angiography in Keratoconus Patients Are Negatively Correlated with Keratoconus Severity

**DOI:** 10.3390/diagnostics14070707

**Published:** 2024-03-27

**Authors:** Martin Dominik Leclaire, Jens Julian Storp, Larissa Lahme, Eliane Luisa Esser, Nicole Eter, Maged Alnawaiseh

**Affiliations:** 1Department of Ophthalmology, University Medical Center Münster, 48149 Münster, Germany; jens.storp@ukmuenster.de (J.J.S.); eliane.esser@ukmuenster.de (E.L.E.); nicole.eter@ukmuenster.de (N.E.); 2Department of Ophthalmology, Klinikum Bielefeld gem. GmbH, 33647 Bielefeld, Germany

**Keywords:** keratoconus, Pentacam, Belin Ambrosio enhanced ectasia display, D score, optical coherence tomography angiography, retinal vessels, choriocapillaris

## Abstract

Keratoconus (KC) is the most common corneal ectasia. Optical coherence tomography angiography (OCT-A) is a relatively new non-invasive imaging technique that allows the visualization and quantification of retinal and choriocapillary blood vessels. The aim of this study is to assess retinal and choriocapillary vessel density (VD) differences between KC patients and healthy controls and to investigate correlations between VD and KC severity. Fifty-two eyes were included in this exploratory study: twenty-six eyes from 26 KC patients and twenty-six eyes from 26 age- and gender-matched healthy controls. All patients underwent Scheimpflug corneal topography with Pentacam, axis lengths measurement and optical coherence tomography angiography (OCT-A). The thinnest spot in corneal pachymetry, maximum K (Kmax) and KC severity indices from the Belin/Ambrósio enhanced ectasia display (BAD) were also assessed. There was a distinct reduction particularly in the retinal VD of the superficial capillary plexus (SCP). Correlation analyses showed strong and moderate negative correlations between the VD in the macular SCP and BAD KC scores and between the SCP VD and Kmax. There was no difference in retinal thickness between the KC and healthy controls. With this study, further evidence for altered VD measurements by OCT-A in KC patients is given. For the first time, we demonstrated negative correlations between BAD KC scores and retinal blood vessel alterations. A major limitation of the study is the relatively small sample size. Since an artefactual reduction of the quantitative OCT-A measurements due to irregular corneal topography in KC must be assumed, it remains to be investigated whether there are also actual changes in the retinal microcirculation in KC.

## 1. Introduction

Keratoconus (KC) is the most common corneal ectasia, with a prevalence of 50–230 cases per 100,000. It is an asymmetric bilateral disorder characterized by a progressive steepening and thinning of the central cornea [1]. It affects mostly young people, and it typically presents in the second decade of life [2]. Keratoconus can lead to impaired visual acuity, and serious complications such as acute corneal hydrops are possible [3]. The pathogenesis is multifactorial. Among the risk factors, which have been described, are environmental factors like eye rubbing and atopy [4]. A positive family history in many cases suggests a genetic component, and several genes have been identified as risk factors [5]. However, the pathophysiology of the disease remains poorly understood. In KC, structural changes in the cornea occur. A thinning of the corneal stroma, an interruption in the Bowman’s membrane and the formation of iron deposits can be observed [3]. Collagen is a key structural protein in the cornea, which plays a crucial role in maintaining its integrity. The expression of collagens I and IV and of other structural proteins is reduced in KC patients [6]. Similar molecular changes happen in connective tissue diseases and a link between Ehlers–Danlos syndrome, Marfan’s syndrome, osteogenesis imperfecta and mitral valve prolapse and KC has been proposed, although, in general, the scientific evidence is weak and few studies exist on this topic [7].

Different grading systems for KC are based on clinical parameters [8]. In the last years, other gradings systems like the Amsler-Krumeich classification were introduced, which consider, in addition to refraction and the presence of corneal scars, keratometry values and corneal thickness [9]. The Scheimpflug-based corneal imaging such as Pentacam (Pentacam, OCULUS GmbH, Wetzlar, Germany) is a method that analyzes both anterior and posterior corneal elevation data and has become established for the diagnosis, grading and follow-up of KC [10,11,12,13]. The Belin/Ambrósio enhanced ectasia display (BAD) has been demonstrated to have a high sensitivity and is especially useful in the detection of early and subclinical KC [14,15]. BAD criteria include the standard deviation (SD) of mean changes in the anterior elevation (df), the SD of mean changes in the posterior elevation (db), the SD of the mean pachymetric progression (dp), the SD of the mean thinnest point thickness (dt), the SD of the mean thinnest point displacement (dy) and the D score, which is calculated by a regression analysis of five determinants [16].

Detection of KC at an early stage can reduce the need for corneal transplantation [17]. It is also essential to recognize early or subclinical KC before performing refractive surgery to avoid inducing iatrogenic ectasia [18]. In everyday clinical routine, the diagnosis and monitoring of KC is generally based on slit lamp examination, on the determination and observation of astigmatism and on Scheimpflug imaging. These diagnostic tools have their limitations, so ways to improve the detection of subclinical forms of KC are being investigated. Recently, it has been demonstrated that the analysis of Scheimpflug imaging, in combination with clinical and demographic parameters, using machine learning algorithms can lead to the improved detection of subclinical KC [19]. OCT imaging is nowadays an integral part of everyday ophthalmological practice, particularly for detecting and monitoring retinal disorders. However, the use of OCT is also increasing in corneal diagnostics [20]. Previous studies have shown that patients with keratoconus display an altered corneal epithelial thickness compared to healthy controls, which can be detected by corneal OCT imaging [21,22].

Optical coherence tomography angiography (OCT-A) is a relatively new imaging technique that provides high-resolution imaging of the retina and choriocapillaris with a resolution in the micrometer range [23,24,25]. OCT-A technology has been described in detail [23]. Briefly, a precise visualization of the blood vessels is achieved by the detection of signal differences caused by the movement of blood cells through multiple, high-resolution scans of a specific area of the retina [23,26,27]. In contrast to conventional dye-based methods such as fluorescein angiography (FA) and indocyanine green angiography (ICG), OCT-A enables non-invasive three-dimensional imaging of the retinal and choriocapillary vasculature [28]. Thus, the retinal vascular plexuses and the choriocapillaris can be visualized and quantified separately [24,29]. With OCT-A, even minor vascular alterations like small quiescent neovascularization, that cannot be detected with conventional dye-based methods can be visualized and monitored [30,31]. In contrast to FA and ICG, OCT-A can be performed in case of dye intolerance, severe renal insufficiency or pregnancy [23,32].

For these reasons, OCT-A is increasingly being used in research, as well as in everyday clinical practice. The most frequently used quantitative OCT-A parameters are the vessel densities (VDs) in the retinal plexuses and in the choriocapillaris and the size of the foveal avascular zone (FAZ) [26,33].

Numerous factors that influence the quantitative measurement parameters of OCT-A, such as axial length, the presence of cardiovascular disease and diabetes mellitus and the presence of cilioretinal vessels and age, have been described [30,34,35,36].

In view of the growing importance of OCT-A in research and clinical application, it is important to detect further possible confounding factors on OCT-A measurements. Moreover, the structural changes in KC and the described association with vascular disease make it desirable to assess retinal and choriocapillary vascular changes in KC patients. Given the importance of the detection of KC and recognition of progression at an early stage, further precise diagnostic options for KC would be desirable. A reduction in VD measured by OCT-A in KC has been described in previous studies, which could make the measurement of retinal VD a potential further diagnostic tool in the detection and monitoring of KC.

In this exploratory study, we aim to evaluate differences in the OCT-A measurements between KC eyes and age-, gender- and axial length-matched healthy controls.

We also analyzed the correlation between retinal and choriocapillary perfusion (as measured by quantitative OCT-A data) and the severity of KC. 

## 2. Materials and Methods

Twenty-six eyes from 26 KC patients and twenty-six eyes from 26 healthy controls were enrolled in this study. The study was approved by the Ethics Committee of the University of Muenster, North Rhine Westphalia, Germany (No. 2015-402-f-S). Written consent about the scientific use of the patient’s data was obtained from all participants. The study adhered to the tenets of the Declaration of Helsinki. Patients with media opacities preventing high-quality imaging, vitreoretinal disease, previous retinal or corneal surgery (including corneal crosslinking), macular edema, glaucoma, arterial hypertension, diabetes, or neurological disease were excluded from the study. All study participants underwent an ophthalmic examination, including an anterior segment examination, binocular fundus examination, corneal topography analysis (details: see below), measurement of the axial length (IOL Master 700, Carl Zeiss Meditec AG, Jena, Germany) and OCT-A imaging (details: see below). Myopia has been described to affect the OCT-A parameters; therefore, patients and participants with axial lengths > 24.5 mm were excluded [37]. Only individuals with normal corneal topographies (as defined by normal K and Kmax values, normal radii, normal corneal thickness and corneal astigmatism ≤ 1 diopter) were included in the healthy control group; normal values were defined based on mean values and standard deviations from previously published data on healthy subjects [38,39,40].

### 2.1. Optical Coherence Tomography Angiography

OCT-A imaging was performed with the AngioVue™ Imaging System (RTVue XR Avanti with AngioVue; Optovue Inc., Fremont, CA, USA). The OCT-A technology has been described before in detail [23]. Briefly, this OCT system with a light source can perform 70,000 scans per second in the A-scan mode centered at 840 nm and with a bandwidth of 45 nm. The axial resolution of this system is 5 μm, and the transverse resolution is 15 μm. The split-spectrum amplitude-decorrelation angiography algorithm is used to create OCT-A data, which were created automatically and displayed and analyzed using Revue software (version 2017.1.0.151, Optovue Inc., Fremont, CA, USA). OCT B-scans obtained progressively at the same location are identical, except for the cell movement within the retinal blood vessels. Thus, blood flow can be visualized by comparing multiple OCT images of the same retinal region by calculation of pixel-by-pixel variations between the scans. The 3 × 3 mm^2^ scans were used for creating OCT-A imaging of the macula, and 4.5 × 4.5 mm^2^ scans were used to acquire pictures of the peripapillary region with the radial peripapillary capillaries (RPCs). Only high-quality OCT-A images were included in this study; scans with artifacts including lines or gaps because of a weak signal strength or motion of the participant were excluded. FAZ boundaries were automatically recognized and marked, and FAZ sizes and FAZ acircularity indices (AI) were calculated by the software. Before data processing and analysis, the automated segmentation and the FAZ marking were verified by an experienced reader. OCT-A imaging of the macula was performed using a 3 × 3 mm scan and OCT-A of the RPCs with a 4.5 × 4.5 mm scan. After checking the automatic segmentation, the retinal thicknesses of the entire retina and of the inner retinal layers (internal limiting membrane to the inner nuclear layer) and of the outer retinal layers (outer plexiform layer to the Bruch’s membrane) were extracted and analyzed. In Figure 1, OCT-A images with the different plexuses and layers and the subregions are displayed exemplarily.

### 2.2. Scheimpflug Corneal Topography

All study participants underwent Scheimpflug corneal topography with Pentacam (Pentacam, OCULUS GmbH, Wetzlar, Germany). Scheimpflug topography was conducted by an experienced examiner under the same conditions, and only examination results without data gaps were used for the study. The Kmax and thinnest spot in corneal pachymetry were extracted and analyzed; furthermore, the BAD measurements were calculated automatically using the Pentacam software (version 1.25, OCULUS GmbH, Wetzlar, Germany).

### 2.3. Statistics

Data were collected in Microsoft Excel 2016. Statistical analyses and the generation of graphs were performed using GraphPad Prism for Windows, Version 10 (GraphPad Software, Boston, MA, USA). 

Statistical advice was obtained in advance of the data analysis (Institute for Biometry and Clinical Research, University of Muenster, Muenster, Germany).

Kolmogorov–Smirnov tests were performed in order to test for normality distribution. Since not all data sets were normally distributed, all *p*-values were evaluated using Mann–Whitney *U* tests for uniformity. Continuous variables are presented as medians and 25% and 75% quartile values.

All *p*-values and confidence limits were intended to be exploratory rather than confirmatory. Therefore, no adjustment for multiplicity was made. Exploratory *p*-values < 0.05 were considered statistically noticeable.

Correlations between two continuous variables were reported as Spearman’s correlation coefficients (r) with 95% confidence intervals.

## 3. Results

### 3.1. Demographic Data and Ocular Data

Demographic data and eye-related data with an emphasis on the KC parameters of the study groups (KC patients and healthy controls) are displayed in Table 1. 

No difference regarding axial lengths (KC group 23.81 [23.30, 24.14] mm, controls 24.04 [23.54, 24.22] years, *p* = 0.8382) was found between the two study groups. 

Corneal parameters such as the corneal astigmatism; thinnest spot in corneal pachymetry; Kmax; d values (df, db, dp, dt and dy) and D score were notably different between the two groups (corneal astigmatism: KC 2.00 [1.35; 3.90] dpt, controls: 0.8 [0.60; 0.98], *p* < 0.0001; corneal pachymetry: KC 482.00 [467.00, 505.30] µm, controls 538.50 [523.30, 558.00] µm, *p* < 0.0001; Kmax: KC 50.95 [47.44; 53.85] dpt, controls 43.60 [42.50, 44.43] dpt, *p* < 0.0001; df: KC 4.42 [3.39, 9.44], controls 0.42 [−0.25, 0.86] µm, *p* < 0.0001, db: KC 3.29 [2.33; 7.14], controls −0.29 [−0.67; −0.08] µm, *p* < 0.0001; dp: KC 5.43 [3.79; 7.17], controls −0.83 [0.25; 1.13] µm, *p* < 0.0001; dt: KC 1.75 [1.07; 2.34], controls −0.07 [−0.79; 0.42] µm, *p* < 0.0001; dy: KC 2.72 [2.29; 3.09], controls 0.56 [0.35; 0.99] µm, *p* < 0.0001; D: KC 5.65 [3.64; 7.21], controls 0.99 [0.40; 1.31] µm, *p* < 0.0001).

### 3.2. Optical Coherence Tomography Angiography Data

A noticeable reduction in the SCP of the macular whole en-face OCT-A and in the parafoveal region of the SCP was detected (SCP whole en face: KC 44.40 [41.58; 47.13], controls 47.40 [46.33; 48.83] µm, *p* = 0.0002; SCP parafovea: 46.70 [44.70; 49.28], controls 50.10 [47.95; 51.93] µm, *p* = 0.0010). There was a trend to the reduction in the foveal region of the SCP (SCP fovea: 17.35 [14.00; 26.15], controls 21.75 [20.25; 25.30] µm, *p* = 0.0719). There were also noticeably reduced VD in the foveal CC VD and the inside disc VD of the RPC OCT-A (CC fovea: KC 71.97 [68.72; 74.02], controls 73.55 [72.01; 75.03], *p* = 0.0310, inside disc RPC: KC 44.30 [39.55; 49.10], controls 49.90 [46.45; 52.28], *p* = 0.0033). No noticeable differences were found for the other parameters.

Data from the OCT-A measurement outcome analysis is presented in Table 2. Additionally, boxplots of the variables with statistically noticeable differences are displayed in Figure 2.

### 3.3. Correlation Analysis

Moderate negative correlations (r ≥ 0.4) were detected for the Kmax and the whole en-face macular SCP VD (r = −0.5836 [−0.7964; −0.2423]), as well as for the df, db, dp and dy values and the whole en-face macular SCP VD (df: r = −0.4702 [−0.7311, −0.0894], db: r = −0.5733 [−0.7907; −0.2276], dp: r = −0.5497 [−0.7774; −0.1946] and dy: r = −0.5734 [−0.7907; −0.2277]). A strong negative correlation (r ≥ 0.6) was found for the D score and the whole en-face macular SCP VD (r = −0.6304 [−0.8220; −0.3107]). Strong negative correlations were also calculated for the Kmax, db value and D score and the foveal SCP VD (Kmax: r = −0.6240 [−0.8185; −0.3012], db: r = −0.6448 [−0−8297; −0.3324] and D: r = −0.6496 [−0.8322; −0.3397]) and moderate correlations for the df, dp and dy values and the foveal SCP VD (df: r = −0.5616 [−0.7814; −0.2112], dp: r = −0.5110 [−0.7552; −0.1424] and dy: r = −0.5103 [−0.7547; −0.1414]). Only weak correlations (r < 0.4) were detected for the control group, except for moderate negative correlations between the db values and the whole en-face macular SCP VD (r = −0.4064 [−0.6921; −0.0105]) and db values and the parafoveal subregion of the SCP VD (r = −0.4435 [−0.7150; −0.0558], *p* = 0.0232). 

Moderate and strong correlation analysis graphs for the KC group are displayed in Figure 3.

### 3.4. Summary of the Results

In summary, there is a noticeably reduced VD, especially in the SCP but also in the foveal region of the choriocapillary VD and in the inside disc region of the RPC in the KC group, compared to healthy controls. KC therefore appears to have a negative influence on the retinal VD, particularly in the macular SCP. Moderate to strong negative correlations were found between the VD of the SCP and the BAD KC severity scores. The possible reasons for the reduced VD depending on the KC severity and the significance of this finding for the use of OCT-A in science and everyday clinical practice are discussed in detail in the following.

## 4. Discussion

In this study, we demonstrate that there are noticeable reductions of the VD, especially in the SCP, in KC patients compared to the healthy controls. The VD was negatively correlated to the Kmax and BAD parameters.

OCT-A is increasingly being used. The unique possibility of rapid and non-invasive imaging of blood vessels and the implication of common diseases like cardiovascular disease, diabetes mellitus and autoimmune diseases on the VD makes OCT-A a promising new technology in diagnosing and monitoring these disorders [36,41,42,43]. It is therefore important that potential confounding factors influencing OCT-A measurement results are known and thus can be considered in the interpretation of measurement results in both scientific applications and in daily clinical practice. Due to the negative correlation of the VD with the severity of KC, which is demonstrated in this study, retinal VD measurements for screening and monitoring the progression of KC could even be a possible further application of OCT-A in the future. Using artificial intelligence (AI) to analyze and interpret OCT-A-generated images can contribute to the enhanced detectability of pathologies and to a more precise quantification of retinal perfusion [44]. In future studies, AI approaches could also potentially assist in more accurately elucidating the impact of KC on OCT-A-based vascular imaging.

Despite the possibilities of OCT-A for high-resolution and detailed visualization of the retinal and choriocapillary vessels, there are technical limitations that are decisive for the image quality and thus for the validity of the OCT-A parameters. The more obvious image errors include motion artifacts, projection artifacts and segmentation errors [45]. It has been shown that artifacts and, in particular, projection artifacts frequently occur in OCT-A examinations in clinical use [46]. The Signal Strength Index (SSI) is an important reference point for the image quality of the OCT-A measurement and for the validity of the quantitative OCT-A parameters [47]. In studies where OCT-A parameters are compared, usually only measurements with a SSI above a defined threshold are used to achieve a certain standard of validity, as it was done in this study [35,42,43,48]. However, a high SSI and the virtual absence of obvious image artifacts do not alone guarantee that unbiased OCT-A measurement parameters are generated. Defocus significantly influences the OCT-A parameters by reducing the visibility of fine capillary vessels, consequently leading to artificially lower measured VD. Interestingly, defocusing appears to have a limited impact on the SSI, suggesting that falsely low measured VD may occur even with acceptable SSI values [49].

There is evidence about the impact of astigmatism on the OCT-A parameters. In a study by Jung et al., it was shown that an induced with-the-rule astigmatism leads to a reduced VD measurement in OCT-A, which the authors attribute to an underestimation of the VD due to a spherical defocus [50]. Another study by Vidal-Oliver et al. demonstrated that, in patients with mild astigmatism, an optical correction of the astigmatism led to a significantly higher mean VD in the OCT-A measurements [51]. A more recent study by the same work group described a significant reduction of the OCT-A VD measurements in eyes with an astigmatism over two diopters. Interestingly, this reduction was more profound in the SCP than in the DCP [52]. Thus, a non-adjustable irregular astigmatism, as found in KC, seems to artefactually influence the OCT-A parameters. 

The question arises as to whether there are actual changes in the retinal and choriocapillary microcirculation in KC patients apart from artifact-induced reductions in VD due to optical defocus. This possibility has been postulated and discussed in detail in previous publications investigating the influence of KC on OCT-A measurements [53,54]. In the following, we discuss the possibility of actual blood vessel changes in KC against the background of the current state of our knowledge.

Although the pathophysiology of KC remains poorly understood until today, changes in collagen structures in KC are eminent. Structural alterations of collagens play a pivotal role in the pathogenesis of connective tissue disorders such as Marfan’s syndrome, Ehlers–Danlos syndrome and osteogenesis imperfecta [55]. These alterations lead to compromised connective tissue strength and abnormal vascular elasticity [56,57,58]. In Marfan’s syndrome, a hereditary connective tissue disorder, there is a deficiency of normal fibrillin-1, an essential component of microfibrils in the extracellular matrix [59]. In the context of Ehlers–Danlos syndrome, a heterogeneous group of disorders characterized by hypermobility and skin fragility, the collagen alterations primarily involve types I, III and V [60]. A link between connective tissue disorders and KC has been proposed: studies have demonstrated that a greater proportion of KC patients shows a hypermobility of joints [61,62]. In patients with Marfan’s syndrome, corneal abnormalities like flattening and astigmatism have been found [63]. Osteogenesis imperfecta is linked to gene mutations responsible for type 1 collagen synthesis [64]. Against the background of the high prevalence of type 1 collagen in the cornea, several studies have investigated the correlation between osteogenesis imperfecta and corneal conditions such as KC. Studies have found a familial clustering of KC cases in families affected by osteogenesis imperfecta and an unusually early onset of KC in these patients [65]. Moreover, the central corneal thickness has been demonstrated to be significantly lower in osteogenesis imperfecta patients [66,67]. KC patients have been demonstrated to have different biophysical characteristics of the skin compared to healthy controls, which is interesting in the context of the frequent skin affection by connective tissue disorders [68].

Connective tissue disorders are linked to vascular alterations. Recent OCT-A studies have demonstrated a reduced retinal VD especially in Marfan patients with systemic vascular disease, and a correlation between VD, FAZ sizes and cardiac function has been proposed [69,70]. Changes in retinal vascular morphology have also been observed in Ehlers–Danlos syndrome patients. In KC, alterations in collagen expression have been described, and it has been hypothesized that elevated levels of matrix metalloproteinases and inflammatory cytokines cause a degradation of collagens [71]. There is also evidence that increased oxidative stress is a contributing factor to degradation of the extracellular matrix in KC [72]. Moreover, collagen type I is disorganized and deformed [73]. The extracellular matrix, containing mainly collagens and elastin, is the main constituent of blood vessel walls [74]. It has been hypothesized that elevated levels of proinflammatory cytokines also affect the cardiovascular system. An association between KC and mitral valve prolapse has been assumed since the 1980s, and a recent meta-analysis, including six studies, indeed found a significant coexisting prevalence between KC and mitral valve prolapse [75]. Back difference elevation measured by Scheimpflug corneal tomography was found to be significantly higher in patients with aortic aneurysm, suggesting a link between aortic aneurysm and KC [76].

The reduced VD in keratoconus, which has been observed several times in the retinal vessels in different studies independently of each other, could indicate possible changes in the structure of the vessel walls. Although there have been few studies on the changes in large vessels, the link between mitral valve prolapse and other connective tissue diseases suggests that there is an association between KC and these diseases and that KC itself may be a generalized connective tissue disease affecting blood vessels. These results indicate a link between cardiovascular disease and KC, and they suggest that there are systemic microvascular changes in KC. However, this assumption remains hypothetical. Further studies of other microvascular systems such as the kidney or skin capillaries using, e.g., capillary microscopy would be desirable in order to detect whether generalized vascular changes occur in KC.

Decreased retinal vessel densities in KC patients have been observed in previous studies investigating KC patients with OCT-A. Wylęgała et al. found significantly reduced vessel densities in KC patients in a recent study, which included 79 KC eyes and 47 healthy eyes from 70 subjects. The reduction was eminent in the full 6 × 6 mm OCT-A, as well as in the subregions (central 1 mm, inner 3 mm without the center 1 mm, outer 6 mm without the inner and center and total area of the ETDRS circle). Moreover, a significantly smaller FAZ area was observed. In this study, the correlation analysis revealed only weak or non-significant correlations between the corneal parameters and VD [53].

Another recent study by Dogan et al. included 32 eyes of 22 KC patients and 24 eyes of 24 age- and sex-matched healthy controls. In this study, there was significantly reduced VD of the macular SCP and DCP and of the inside disc RPC VD, but in contrast to Wylęgała et al., no difference in FAZ sizes was observed in the KC patients [53,54]. Moreover, the choriocapillary flow area was significantly higher in the KC patients [54]. A further study by Pierro et al. included 32 eyes with KC and 32 age- and axial length-matched control eyes with mostly early-stage KC [77]. In this study, a significant reduction of the peripapillary and the macular SCP VD was detected.

Zırtıloğlu also demonstrated reduced VD mainly in the SPC. In this study, a negative correlation between the KC stage (as defined by the Amsler-Krumeich classification staging system) was found [78].

It is known that the retinal VD, especially in the SCP, correlates with the retinal thickness [79,80,81,82,83]. In numerous conditions like axial myopia, diabetes mellitus, glaucoma and Alzheimer’s disease, a reduced VD has been reported, which is correlated with a reduction in retinal thickness [84,85,86,87,88]. There are differing results in the literature regarding retinal thickness in KC measured by OCT. Some studies have reported no differences in macular retinal thicknesses between KC and healthy controls [89,90], which also applies to the OCT-A study by Wylęgała et al., where the central macular thickness was also assessed [53]. Other studies have shown reduced retinal thicknesses, both in terms of macular thickness, as well as retinal nerve fiber layer thickness and ganglion cell layer thickness [91,92]. Further studies have reported a higher retinal and macular thickness in KC [93,94]. One study demonstrated an increased thickness of the inner nuclear layer with a reduced thickness of the outer retinal layers [95]. Changes in the retinal thickness in KC are often interpreted in the literature as structural changes in the posterior segment parameters as an expression of retinal plasticity [93,96]. In this study, we observed no difference in the average retinal thickness across the entire 3 × 3 mm OCT-A scan, as well as in the subregions (foveal and parafoveal). This applies to both the overall thickness and the separately analyzed thickness of the inner retina and the outer retina. This could be interpreted as an indication that a defocus-related optical error due to uncorrectable irregular astigmatism is the cause of the reduced VD in the SCP rather than actual retinal vascular changes.

It is noticeable that the reduction in VD occurs primarily in the SCP and not in the whole en-face RPC or the whole en-face CC. The reasons for this are speculative. It could be hypothesized that the VD of the RPC and the CC is less validly measurable and therefore provides less conclusive results. However, previous studies have shown that the reproducibility of the RPC and CC VD is good [97,98,99,100,101]. If this is the case, a pure defocus and a purely artificial reduction of the VD due to uncorrected irregular astigmatism would not be the only sufficient explanation for the reduced VD in the SCP, as a reduction of all retinal vascular plexuses, the CC VD and the RPC VD would then have to be assumed. Further research is needed to answer the question of the reasons for the reduced VD mainly of the SCP in KC patients. A possible approach for further studies would be to conduct VD measurements in KC patients wearing contact lenses that compensate for the irregular astigmatic refractive errors. This was not possible in this study, as the patients had to adhere to contact lens cessation for the validity of the Scheimpflug measurements.

In summary, there is a potential association between keratoconus (KC) and vascular alterations. However, due to the confounding influence of defocus, caused by irregular and not fully correctable astigmatism in KC, the described OCT-A changes do not provide definite information about a genuine VD reduction and alterations in the retinal perfusion in KC.

In our study, we found a noticeable negative correlation between the whole en-face and parafoveal VD in the SCP and main KC severity factors (Kmax and BAD parameters). This suggests a relationship between the severity of KC and the reduced VD in the SCP.

The reduced retinal VD is a result that has been found by previous OCT-A studies on KC patients by Wylęgała et al., Dogan et al., Pierro et al. and Zırtıloğlu et al., as well [53,54,77,78]. However, the results regarding the VD in the DCP, the CC VD and the FAZ size are not consistent between our study and the two previous OCT-A studies on KC. In contrast to our results, Wylęgała et al. reported a significantly reduced FAZ size in KC [53], Dogan et al. demonstrated a significant reduction in DCP VD and an increased VD in the whole en-face CC VD [54] and Zırtıloğlu et al. also showed reduced VD in the whole DCP and the whole RPC OCT-A VD measurements [78]. These differences may be due to the following reasons: First, Wylęgała et al. used a different OCT device and different software to obtain the OCT-A images (Zeiss Cirrus 500, Carl Zeiss Meditec AG, Jena, Germany and Angioplex software, version 11) [53]. It is known that the results of the different systems are only comparable with each other to a limited extent. Even in young, healthy subjects, there is little agreement in the measurements among the different devices, and thus, their results are not interchangeable [102]. This applies to VD, as well as to FAZ measurements [103,104]. Second, even though the results from different studies are incongruent, some studies reported a higher interpersonal variation of VD measurements in the DCP than in the SCP, which would make the overall evaluability and comparability of the DCP measurements less reliable [82,105]. Moreover, divergent results between the SCP and DCP have been reported in OCT-A studies of various conditions [42,106,107]. It is well known that axial myopia negatively influences OCT-A-based VD measurements [37]. The average axial length in KC eyes has been reported to be longer than in emmetropic eyes [64], which is why a possible bias should be excluded in OCT-A studies with KC patients using axial length measurements. However, in the studies by Wylęgała et al. and by Zırtıloğlu et al., no axial length measurements were conducted [53,78].

Further, larger studies with KC patients and healthy controls would be desirable to deeper analyze the incongruities between the different studies.

## 5. Limitations

One limitation of the study is the relatively small number of patients, which is because patients with previous crosslinking or other media opacities such as corneal scars after corneal hydrops were excluded, thus limiting the number of available subjects in a tertiary medical center where the proportion of complicated KC cases was higher than in primary care. In addition, only OCT-A images of good quality were used, and patients with axial lengths > 24.5 mm were excluded, which further limited the number of eyes to be included.

It is likely that optical errors, such as defocus due to the irregular corneal surface and astigmatism in KC, influenced the OCT-A measurements. As a result, it cannot be stated whether actual vascular alterations in KC patients exist, even though this has been assumed in earlier studies. The anticipated difference in refractive errors between the KC group and the healthy subjects represents another limitation of the study.

## 6. Conclusions

In summary, our results demonstrate that retinal VD measured by OCT-A is noticeably reduced in KC patients. We found a negative correlation between important parameters for the grading of KC severity and the VD. 

As defocus caused by not correctable astigmatism significantly impacts the validity of OCT-A measurements and leads to artefactually reduced VD measurements, no clear conclusion can be drawn regarding actual changes in retinal microcirculation in KC. More scientific evidence regarding the systemic implications of keratoconus, particularly on the vascular system and especially on the microcirculation, would be desirable in order to better understand the connections between KC and vascular disease.

Considering the growing importance of OCT-A and the frequency of KC, it is important to identify KC as an influencing factor on OCT-A based measurements and to be able to take it into account accordingly, for example, in the interpretation of VD measurements or as an exclusion criterion for healthy control groups in comparative studies.

## Figures and Tables

**Figure 1 diagnostics-14-00707-f001:**
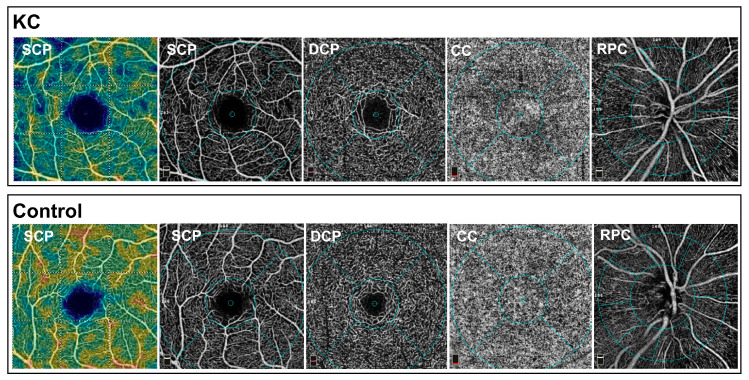
Exemplary images of the optical coherence tomography angiography (OCT-A) vessel density (VD) measurements. Exemplary participants from the KC group (**top**) and the control group (**bottom**). Displayed are from left to right: heat map of the VD of the superficial capillary plexus (SCP), vessel visualization of the SCP, vessel visualization of the deep capillary plexus (DCP), the choriocapillaris (CC) and the radial peripapillary capillaries (RPCs). The blue circles indicate the subregions. Note the reduced VD in the SCP heat map in the KC eye compared to the healthy control.

**Figure 2 diagnostics-14-00707-f002:**
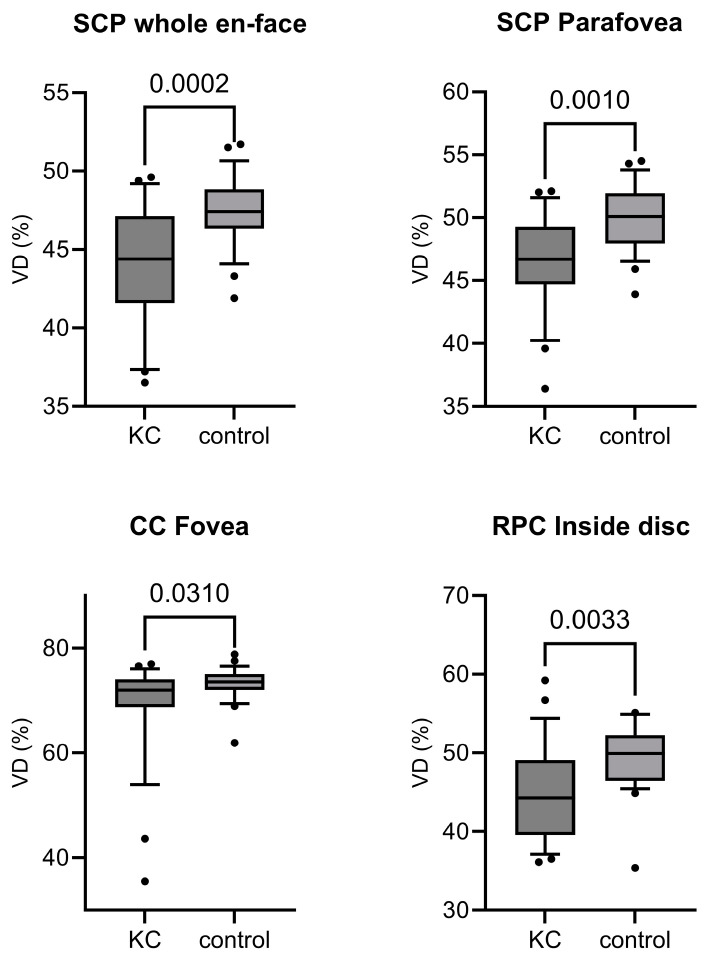
Visualization of the variables with a statistically noticeable difference. Displayed are the measurement outcomes of the vessel densities (VD) obtained by optical coherence tomography angiography (OCT-A). (**Top left**) Superficial capillary plexus (SCP) whole en face OCT-A values, (**top right**) SCP OCT-A values of the parafoveal region, (**down left**) foveal region OCT-A values of the choriocapillary VD and (**bottom right**) radial peripapillary plexus (RPC) OCT-A VD in the inside disc region. Box plots represent medians and 25th and 75th percentiles, and whiskers represent 10th to 90th percentiles. *p*-values are shown above the brackets (Mann–Whitney *U* tests). KC: keratoconus group, control: healthy controls (*n* = 26 for each group).

**Figure 3 diagnostics-14-00707-f003:**
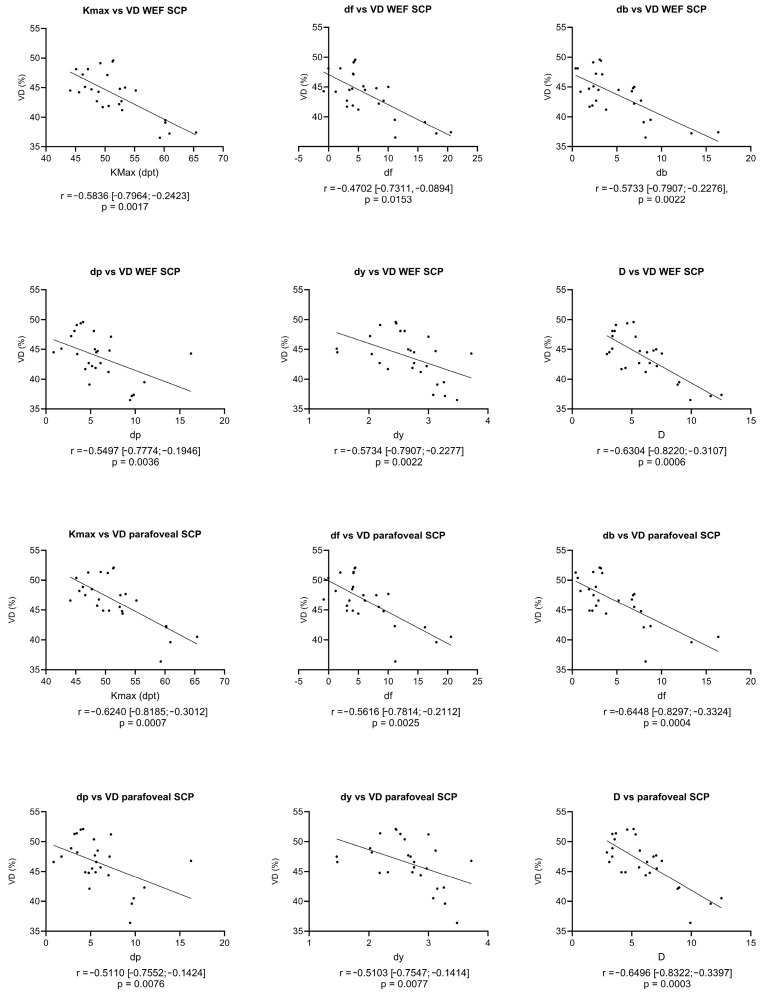
Correlations between whole en-face (WEF) and parafoveal vessel densities (VDs, in %) of the superficial capillary plexus (SCP) and KC severity parameters. Linear regression curves are displayed, and the correlation was calculated as the Spearman correlation coefficient (r) in parentheses: 95% confidence intervals. df: standard deviation (SD) of mean changes in the anterior elevation, db: SD of mean changes in the posterior elevation, dp: SD of the mean pachymetric progression, dt: SD of the mean thinnest point thickness, dy: SD of the mean thinnest point displacement, D: D score, dpt: diopters and Kmax: maximum keratometry.

**Table 1 diagnostics-14-00707-t001:** Demographic and ocular data.

	KC (*n* = 26)	Controls (*n* = 26)	*p*
*n*	26	26	n.a.
gender	4/22	4/22	n.a.
age (y)	25.48 [20.48, 34.78]	26.05 [23.68, 29.83]	0.8382
axial lengths (mm)	23.81 [23.30, 24.14]	24.04 [23.54, 24.22]	0.3465
corneal astigmatism (dpt)	2.00 [1.35; 3.90]	0.8 [0.60; 0.98]	**<0.0001**
thinnest spot corneal pachymetry (µm)	482.00 [467.00, 505.30]	538.50 [523.30, 558.00]	**<0.0001**
Kmax (dpt)	50.95 [47.55; 53.85]	43.60 [42.50, 44.43]	**<0.0001**
df	4.42 [3.39; 9.44]	0.42 [−0.25; 0.86]	**<0.0001**
db	3.29 [2.33; 7.14]	−0.29 [−0.67; −0.08]	**<0.0001**
dp	5.43 [3.79; 7.17]	0.83 [0.25; 1.13]	**<0.0001**
dt	1.75 [1.07; 2.34]	−0.07 [−0.79; 0.42]	**<0.0001**
dy	2.72 [2.29; 3.09]	0.56 [0.35; 0.99]	**<0.0001**
D	5.65 [3.64; 7.21]	0.99 [0.40; 1.31]	**<0.0001**

Left: Keratoconus (KC) patients, right: healthy controls. Displayed are absolute numbers and medians with 25th and 75th percentiles (in parentheses). corneal pachymetry: thinnest spot in corneal pachymetry measurement, df: standard deviation (SD) of mean changes in the anterior elevation, db: SD of mean changes in the posterior elevation, dp: SD of the mean pachymetric progression, dt: SD of the mean thinnest point thickness, dy: SD of the mean thinnest point displacement, D: D score, dpt: diopters, Kmax: maximum keratometry, mm: millimeters, µm: micrometers, *n*: number, n.a.: not applicable and y: years. Bold: *p*-values < 0.05.

**Table 2 diagnostics-14-00707-t002:** Optical coherence tomography angiography measurement results.

	KC (*n* = 26)	Controls (*n* = 26)	*p*
SCP (VD)			
whole en face	44.40 [41.58; 47.13]	47.40 [46.33; 48.83]	**0.0002**
Fovea	17.35 [14.00; 26.15]	21.75 [20.25; 25.30]	0.0719
Parafoveal	46.70 [44.70; 49.28]	50.10 [47.95; 51.93]	**0.0010**
DCP (VD)			
whole en face	50.60 [46.55; 54.25]	50.70 [48.20; 53.40]	0.5871
Fovea	36.10 [31.55; 42.53]	38.35 [34.73; 42.58]	0.3016
Parafoveal	53.35 [48.85; 55.75]	52.60 [50.20; 54.65]	0.9819
CC (VD)			
whole en face	73.73 [71.41; 75.21]	73.15 [71.74; 74.70]	0.7682
Fovea	71.97 [68.72; 74.02]	73.55 [72.01; 75.03]	**0.0310**
Parafoveal	73.79 [71.27; 76.10]	72.67 [71.51; 74.62]	0.2245
RPC (VD)			
whole en face	47.70 [46.00; 50.90]	47.65 [46.00; 50.10]	0.6359
inside disc	44.30 [39.55; 49.10]	49.90 [46.45; 52.28]	**0.0033**
Peripapillary	51.10 [49.35; 53.80]	50.45 [48.88; 53.15]	0.4571
CRT (µm)	264.00 [248.50; 276.30]	264.5 [254.00; 270.30]	0.7889
FAZ (mm^2^)	0.23 [0.14; 0.30]	0.20 [0.14; 0.25]	0.2590
AI	0.13 [0.11; 0.16]	0.12 [0.11; 0.16]	0.6388
RT: all layers (µm)			
whole en face	334.80 [320,10; 339.50]	332.70 [323.20; 343.20]	0.4593
Fovea	273.60 [255.30; 288.90]	275.60 [266.60; 281.10]	0.7039
Parafoveal	343.40 [330.50; 349.20]	341.30 [332.70; 352.80]	0.9442
RT: inner layers (µm)			
whole en face	172.60 [167.90; 181.00]	175.60 [170.90; 183.30]	0.9283
Fovea	89.10 [76.40; 106.90]	94.40 [91.30; 106.40]	0.7490
Parafoveal	183.50 [177.90; 191.20]	184.90 [180.70; 191.50]	0.7642
RT: outer layers (µm)			
whole en face	157.10 [152.20; 162.90]	156.60 [150.90; 160.80]	0.6384
Fovea	180.30 [174.50; 189.00]	182.30 [170.80; 186.70]	0.6672
Parafoveal	157.60 [152.10; 163.40]	156.20 [149.80; 161.40]	0.7279

Left: Keratoconus (KC) patients, right: healthy controls. Displayed are absolute numbers and medians with 25th and 75th percentiles (in parentheses). AI: acircularity index, CC: choriocapillaris, CRT: central retinal thickness, DCP: deep retinal capillary plexus, FAZ: foveal avascular zone size, RPCs: radial peripapillary capillaries, RT: retinal thickness, SCP: superficial retinal capillary plexus, inner retinal layers: internal limiting membrane to the inner nuclear layer and outer retinal layers: outer plexiform layer to the Bruch’s membrane. Bold: *p*-values < 0.05.

## Data Availability

Data are contained within the article.

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
