# Peer review of "Reduced Retinal Blood Vessel Densities Measured by Optical Coherence Tomography Angiography in Keratoconus Patients Are Negatively Correlated with Keratoconus Severity"

_diagnostics, 2024, doi:10.3390/diagnostics14070707_

Round 1
Reviewer 1 Report (Previous Reviewer 1)
Comments and Suggestions for Authors
Although, the manuscript was substantially revised it still contains many unproven statements on retinal vascular changes in keratoconus. I’m personally sure that the only way you can interpret your results is to say about artefactual changes of OCTA imaging which was clearly concluded from previous studies. It is also quite clear that irregular astigmatism obviously would result in more severe OCTA imaging disturbance due to higher degree of high-order aberrations.
As I believe, authors know that the loss of retinal vessel density is usually associated with the loss of retinal thickness as we can see in many other conditions. Were any notable volumetric characteristics of the neurosensory retina and its separate layers seen in your KC patients? Please provide inner and outer retina thickness in your KC cohort and healthy controls with astigmatism.
Common abbreviation for fluorescein angiography is FA.
“Despite a possible spherical defocus due to the corneal changes in KC, which may influence the optical imaging of the blood vessels in OCT-A” I do not understand why authors reject absolutely simple idea that defocus and optical aberrations (including astigmatism) ARE ACTUALLY (not “MAY”) affect OCTA imaging. Please perform your OCTA examinations without adjusting (including focusing) OCTA machine… will you acquire adequate images in non-emmetropic eyes? I doubt… The same is for CK, but you cannot adjust irregular astigmatism.
May personal opinion remains the same: the results of your study may be interpreted only as artefactual effects of OCTA imaging of normal retinal microcirculation in CK patients.
Author Response
Although, the manuscript was substantially revised it still contains many unproven statements on retinal vascular changes in keratoconus. I’m personally sure that the only way you can interpret your results is to say about artefactual changes of OCTA imaging which was clearly concluded from previous studies. It is also quite clear that irregular astigmatism obviously would result in more severe OCTA imaging disturbance due to higher degree of high-order aberrations.
Previous publications about OCT-A and KC have all intensively discussed the possibilities of actual vascular changes and paid little attention to the aspect of optical errors caused by uncorrectable irregular astigmatism. We have adapted the text of the manuscript again according to your suggestions and emphasized that the OCT-A results cannot be used to draw clear conclusions about actual vascular changes in KC and that an artifact-related reduction of the measurement parameters must be assumed.
Thank you for your expertise, which has significantly improved the quality of our manuscript and sets it apart from previous publications.
We have made the following extensive text changes to the manuscript:
Abstract:
We added the following sentecens:
“There was no difference in retinal thicknesses between KC and healthy controls.”
“Since an artefactual reduction of the quantitative OCT-A measurements due to irregular corneal topography in KC must be assumed, it remains to be investigated whether there are also actual changes in the retinal microcirculation in KC.
instead of
“Also, it remains unclear whether the reduced VD is due to KC-related optical errors or to actual changes in the microcirculation in KC.”
Discussion:
the following passage was added:
“Despite the possibilities of OCT-A for high-resolution and detailed visualization of the retinal and choriocapillary vessels, there are technical limitations that are decisive for the image quality and thus for the validity of the OCT-A parameters. The more obvious image errors include motion artifacts, projection artifacts and segmentation errors [45]. It has been shown that artifacts and in particular projection artifacts frequently occur in OCT-A examinations in clinical use [46]. The Signal Strength Index (SSI) is an important reference point for the image quality of the OCT-A measurement and for the validity of the quantitative OCT-A parameters [47]. In studies where OCT-A pa-rameters are compared, usually only measurements with an SSI above a defined threshold are used to achieve a certain standard of validity [35,42,43,48]. However, a high SSI and the virtual absence of obvious image artifacts do not alone guarantee that unbiased OCT-A measurement parameters are generated. Defocus significantly influences OCT-A parameters by reducing the visibility of fine capillary vessels, consequently leading to artificially lower measured VD. Interestingly, defocusing appears to have a limited impact on the SSI, suggesting that falsely low measured VD may occur even with acceptable SSI values [49].”
We changed the following passages:
“There is evidence about the impact of astigmatism on OCT-A parameters. In a study by Jung et al., it was shown that an induced with-the-rule astigmatism leads to a reduced VD measurement in OCT-A, which the authors attribute to an underestimation of the VD due to a spherical defocus [50]. Another study by Vidal-Oliver et al. demonstrated that in patients with mild astigmatism, an optical correction of the astigmatism led to significantly higher mean VD in OCT-A measurements [51]. A more recent study by the same work group described a significant reduction of OCT-A VD measurements in eyes with an astigmatism over two diopters. Interestingly, this reduction was more profound in the SCP than in the DCP [52]. Thus, a non-adjustable irregular astigmatism, as found in KC, seems to artefactually influence the OCT-A parameters.
The question arises as to whether there are actual changes in the retinal and choriocapillary microcirculation in KC patients apart from artifact-induced reductions in VD due to optical defocus. This possibility has also been postulated and discussed in detail in previous publications investigating the influence of KC on OCT-A measurements [53,54]. In the following, we discuss the possibility of actual blood vessel changes in KC against the background of the current state of knowledge.”
instead of
“The question arises as to whether optical changes caused by KC influence OCT-A measurements. In a study by Jung et al., it was shown that an induced with-the-rule astigmatism leads to a reduced VD measurement in OCT-A, which the authors attribute to an underestimation of the VD due to a spherical defocus [50]. Another study by Vidal-Oliver et al. demonstrated that in patients with mild astigmatism, an optical correction of the astigmatism led to significantly higher mean VD in OCT-A measurements [51]. A more recent study by the same work group described a significant reduction of OCT-A VD measurements in eyes with an astigmatism over two diopters. Interestingly, this reduction was more profound in the SCP than in the DCP [52]. These results, however, refer to regular astigmatism, which can only be compared to a limited extent with irregular astigmatism in KC.
Nevertheless, an influence of a defocus due to the complex refractive power changes of a cornea altered by the KC on the OCT-A measurement parameters is possible, especially against the background of the negative influence of astigmatisms on the VD described in the mentioned studies. A possible approach for further research would be to conduct VD measurements in KC patients wearing contact lenses that compensate for the irregular astigmatic refractive errors. This was not possible in this study, as patients had to adhere to contact lens cessation for the validity of Scheimpflug measurements.”
“The question arises as to whether there are actual changes in the retinal and choriocapillary microcirculation in KC patients apart from artifact-induced reductions in VD due to optical defocus. This possibility has also been postulated and discussed in detail in previous publications investigating the influence of KC on OCT-A measurements [53,54]. In the following, we discuss the possibility of actual blood vessel changes in KC against the background of the current state of knowledge.”
instead of
“This possibility has also been postulated and discussed in detail in previous publications investigating the influence of KC on OCT-A measurements [53,54].”
“These results indicate a link between cardiovascular disease and KC, and they suggest that there are systemic microvascular changes in KC. However, this assumption re-mains hypothetical. Further studies of other microvascular systems such as the kidney or skin capillaries using e.g. capillary microscopy would be desirable in order to detect whether generalized vascular changes occur in KC.”
Instead of
“These results indicate a link between cardiovascular disease and KC, and they suggest that there are systemic microvascular changes in KC. Further studies of other microvascular systems such as the kidney or skin capillaries using e.g. capillary microscopy would be desirable in order to detect whether generalized vascular changes occur in KC.”
We added this sentence to the discussion:
“In summary, there is a potential association between keratoconus (KC) and vascular alterations. However, due to the confounding influence of defocus, caused by irregular and not fully correctable astigmatism in KC, the described OCT-A changes do not provide definite information about a genuine VD reduction and alterations in the retinal perfusion in KC.”
The following passages were removed:
“Despite a possible spherical defocus due to the corneal changes in KC, which may influence the optical imaging of the blood vessels in OCT-A, the results of our study and those of the other mentioned OCT-A studies with KC patients could also indicate noticeable changes in the retinal microcirculation compared to healthy controls [53,54,78,79].” (Deleted)
“In the following, we discuss the possible reasons for the differences between retinal and choriocapillary VD in keratoconus.
The retina is supplied with blood from two distinct vascular systems, the retinal blood vessels, and the choroid. The retinal blood vessels are derived from the central retinal artery whereas the choroid is derived from the ciliary arteries [80]. The choroidal and the retinal vasculature differ in terms of blood flow, autoregulation, and oxygen saturation of their blood [81,82].
Gutierrez-Bonet demonstrated a significantly enlarged choroidal thickness measured by swept source (SS-)OCT especially in young KC patients compared to healthy controls [83]. In another large study with 97 KC eyes and 145 eyes of healthy controls, the same work group showed an age-adjusted increased vascularity index in KC patients in SS-OCT [84]. The authors propose a possible inflammatory infiltration and vascular dilation as causes of the increased choroidal thickness. These results suggest structural changes in the choroid in KC. The CC is an approximately 10 µm thin layer at the inner part of the choroid [85]. OCTA provides a layer visualization of the CC separate from the underlying rest of the choroid and separate from the retinal vascular plexus [24]. In this study, retinal SCP VD was significantly reduced, whereas there was no difference in the whole-en face CC VD between KC patients and healthy controls. However, in the foveal subregion of the CC OCT-A, VD was noticeably reduced as compared to controls. The reasons for choroidal changes in KC patients, which can be assumed based on the mentioned OCT and OCT-A studies, are speculative and further studies including histopathological examinations of the choroid in KC patients would be desirable.
Considering the current knowledge, it could be speculated that KC leads to structural changes with resulting decreased VD the retinal vessels. These changes do not seem to be detectable everywhere in the CC. It is possible that autoregulatory mechanisms of the retinal circulations play a role to compensate for the increased choroidal vascular index. Another possible explanation would be structural variances between retinal and choroidal vessels.” (Deleted)
Limitations
It is likely that optical errors, such as defocus due to the irregular corneal surface and astigmatism in KC, influence the OCT-A measurements. As a result, it cannot be stated whether actual vascular alterations in KC patients exist, even though this was assumed in earlier studies. The anticipated difference in refractive errors between the KC group and the healthy subjects represent another limitation of the study.
Conclusions
The following sentences were added:
“As defocus caused by not correctable astigmatism significantly impacts validity of OCT-A measurements and leads to artefactually reduced VD measurements, no clear conclusion can be drawn regarding actual changes in retinal microcirculation in KC. More scientific evidence regarding the systemic implications of keratoconus, particu-larly on the vascular system and especially on the microcirculation, would be desirable in order to better understand the connections between KC and vascular disease.”
As I believe, authors know that the loss of retinal vessel density is usually associated with the loss of retinal thickness as we can see in many other conditions. Were any notable volumetric characteristics of the neurosensory retina and its separate layers seen in your KC patients? Please provide inner and outer retina thickness in your KC cohort and healthy controls with astigmatism.
Thank you for this important suggestion. We extracted the retinal thickness data from the OCT-A measurements and added them to Table 1.
We also added the following passage to the discussion:
“It is known that the retinal VD, especially in the SCP, correlates with retinal thickness [79–83]. In numerous conditions like axial myopia, Diabetes mellitus, Glaucoma and Alzheimer’s disease a reduced VD has been reported, which is correlated with a reduction in retinal thickness [84–88]. There are differing results in the literature regarding retinal thickness in KC measured by OCT. Some studies report no differences in macular retinal thicknesses between KC and healthy controls [89,90], which also applies to the OCT-A study by WylÄ™gaÅ‚a et al., where the central macular thickness was also assessed [53]. Other studies showed reduced retinal thicknesses, both in terms of macular thickness as well as retinal nerve fiber layer thickness and ganglion cell layer thickness [91,92]. Further studies report a higher retinal and macular thickness in KC [93,94]. One study demonstrated an increased thickness of the inner nuclear layer with reduced thickness of the outer retinal layers [95]. Changes in retinal thickness in KC are often interpreted in the literature as structural changes in the posterior segment parameters as an expression of retinal plasticity [93,96].
In this study, we observed no difference in the average retinal thickness across the en-tire 3×3 mm OCT-A scan, as well as in the subregions (foveal and parafoveal). This ap-plies to both the overall thickness and the separately analyzed thickness of the inner retina and the outer retina. This could be interpreted as an indication that a defocus-related optical error due to the uncorrectable irregular astigmatism is the cause of the reduced VD in the SCP rather than actual retinal vascular changes.”
Common abbreviation for fluorescein angiography is FA.
Abbreviation for fluorescein angiography was changed from “FAG” to “FA”.
“Despite a possible spherical defocus due to the corneal changes in KC, which may influence the optical imaging of the blood vessels in OCT-A” I do not understand why authors reject absolutely simple idea that defocus and optical aberrations (including astigmatism) ARE ACTUALLY (not “MAY”) affect OCTA imaging. Please perform your OCTA examinations without adjusting (including focusing) OCTA machine… will you acquire adequate images in non-emmetropic eyes? I doubt… The same is for CK, but you cannot adjust irregular astigmatism.
May personal opinion remains the same: the results of your study may be interpreted only as artefactual effects of OCTA imaging of normal retinal microcirculation in CK patients.
We added the above-mentioned passages about the limitation that a defocus influences VD measurements and therefore, no conclusion about actual perfusion changes in KC can be made.
We would like to thank you once again for reviewing the manuscript and for drawing attention to the influence of astigmatism-induced defocus. Your important comments have led to a much more differentiated discussion than in the original manuscript.

Reviewer 2 Report (Previous Reviewer 2)
Comments and Suggestions for Authors
The authors have satisfactorily responded to the reviewers' suggestions in the revised text.
Comments on the Quality of English LanguageMinor edits are needed, which can be addressed during final editorial proofing.
Author Response
The authors have satisfactorily responded to the reviewers' suggestions in the revised text.
Thank you for your comments and suggestions, which have clearly improved the quality of the manuscript.
Reviewer 3 Report (Previous Reviewer 3)
Comments and Suggestions for Authors
Authors performed prominent improvement of the manuscript.
But I still recommend to include to Conclusions only statments, but not discussion. Better to remove: To the best of our knowledge, in 511 this study we analyzed the correlation between OCT-A based VD measurements and the 512 BAD D score for the first time, whereas in a previous publication a negative correlation 513 between VD measurements and the older Amsler-Krumeich classification was analyzed 514 [72]
Please change number of Conclusion to 6.
Author Response
Authors performed prominent improvement of the manuscript.
Thank you very much for your input and for reviewing again!
But I still recommend to include to Conclusions only statments, but not discussion. Better to remove: To the best of our knowledge, in 511 this study we analyzed the correlation between OCT-A based VD measurements and the 512 BAD D score for the first time, whereas in a previous publication a negative correlation 513 between VD measurements and the older Amsler-Krumeich classification was analyzed 514 [72]
We removed this passage from the conclusions section in order to reduce it to statements.
Please change number of Conclusion to 6.
Thank you for pointing out this typo, we have corrected the number.
Round 2
Reviewer 1 Report (Previous Reviewer 1)
Comments and Suggestions for Authors
My comments were adequately addressed. Thanks to the authors.
This manuscript is a resubmission of an earlier submission. The following is a list of the peer review reports and author responses from that submission.
Round 1
Reviewer 1 Report
Comments and Suggestions for Authors
I appreciated author’s effort in improving their manuscript.
However, I personally believe that the authors came to the wrong conclusion. Only the study incorporated measurements with astigmatism correction may answer this question.
“depicted association of keratoconus with connective tissue disorders, and in turn their association with vascular diseases, likely indicates actual changes in retinal perfusion. Taken together, the results of this study could suggest that KC is not a disease limited to the cornea alone, but that the architecture of the retinal and choriocapillary blood vessels is also altered, which could indicate generalized collagen changes in KC”
I recommend to the authors to look at the recent paper of Vidal-Oliver [Lourdes Vidal-Oliver, Roberto Gallego-Pinazo, Rosa Dolz-Marco; Astigmatism Influences Quantitative and Qualitative Analysis in Optical Coherence Tomography Angiography Imaging. Trans. Vis. Sci. Tech. 2024;13(1):10.], which again showed that astigmatism more than 2.0 D reduces VD density. This paper also showed that SCP VD is more susceptible to this effect... By the way, I failed to find the astigmatism power in the revised manuscript.
Reviewer 2 Report
Comments and Suggestions for Authors
From a reviewer’s point of view, overall, the authors have done a clear study that employs optical coherence tomography angiography (OCT-A), a well-established imaging technique, to assess retinal and choriocapillary vessel density (VD) in keratoconus (KC) patients and healthy controls. This methodological choice aligns with current standards in ophthalmic research. The inclusion of age- and gender-matched healthy controls enhances the scientific rigor of the study by controlling for potential confounding variables. The findings of reduced retinal vessel density in KC patients compared to controls are consistent with previous research, providing support for the scientific soundness of the study.
Comments:
1. Abstract:
a. Authors have used OCTA for VD and KC measurements. The abstract accurately summarizes the study objectives, methodology, and key findings, providing a clear overview of the research conducted. However, limitations in a study, like sample size and variables used for statistical analysis, can be briefly highlighted in the abstract.
2. Introduction:
a. The authors have given a clear background for the importance of studying VD and KC, but the key part of their study is the use of OCTA for their measurements. However, the background of OCTA is very brief. It is much too abstract. The working principle, previous works, and applications of OCT in EYS diagnosis are not given. This makes it very difficult for readers to understand the potential advantages of using OCTA. This MUST be addressed in the introduction section.
b. Authors have briefly addressed the current widely used methods used for KC VD measurements. But this is not enough. The authors must elaborate on the limitations of current techniques and comparatively how OCTA will be a valuable measurement method.
3. Materials and Methods:
a. Information about axial and lateral resolutions of the commercial OCTA system used in their study is not mentioned.
4. Results:
a. The results sections are brief. It is to be noted that these are discussed extensively in the discussion section. Still, the major interpretations of the result measurements should be highlighted in the result section and not just the obtained statistical values. Without this, the result correlations will be difficult to interpret and grasp to their full extent.
5. Discussion:
a. It is important for the authors to acknowledge potential limitations of the study, such as the relatively small sample size and the possibility of confounding variables influencing the observed associations. Additionally, the generalizability of the findings may be limited by the inclusion criteria and characteristics of the study population. Clear addressing with detailed explanations of limitations would strengthen the scientific integrity of the article.
b. Can the authors include more detail about the different future uses for implementing their study results in demonstrating how retinal VD measured by OCT-A, which showed a noticeable reduction in KC patients, can be useful in research and clinical scenarios?
Also, can the authors include the use of Learning and other Machine learning methods that are currently being explored and reported by researchers and doctors alike for a better understanding of different imaging diagnostics and for early diagnostics? Also, even a short mention of how AI-assisted image analysis and machine learning can be expected to further advance the understanding of VD in KC patients.
Comments on the Quality of English LanguageThe article is well written, but the flow of the article can be improved, and it needs minor to moderate changes with the English Language. Given the authors address all comments, the manuscript can be considered for acceptance.
Reviewer 3 Report
Comments and Suggestions for Authors
Text in rows 148-149 (In order to pre- 148 vent a bias [19], individuals with long axial lengths (> 24.5 mm) were excluded .) better to exclude, becaus it duplicates text in Materials (rows 85-86)
Limitation of the study (rows 437-443) better to be moved before Conclusions.
I reccomend to include to Conclusions only statments, but not discussion.
